# 5G Radiofrequency Exposure Reduces *PRDM16* and *C/EBP β* mRNA Expression, Two Key Biomarkers for Brown Adipogenesis

**DOI:** 10.3390/ijms26062792

**Published:** 2025-03-20

**Authors:** Chandreshwar Seewooruttun, Bélir Bouguila, Aurélie Corona, Stéphane Delanaud, Raphaël Bodin, Véronique Bach, Rachel Desailloud, Amandine Pelletier

**Affiliations:** 1PériTox (UMR I_01), UPJV/INERIS, University of Picardy Jules Verne, CURS, Chemin du Thil, 80025 Amiens, France; chandreshwar.seewooruttun@u-picardie.fr (C.S.); belir.bouguila@u-picardie.fr (B.B.); aurelie.corona@u-picardie.fr (A.C.); stephane.delanaud@u-picardie.fr (S.D.); veronique.bach@u-picardie.fr (V.B.); desailloud.rachel@chu-amiens.fr (R.D.); 2PériTox (UMR I_01), INERIS/UPJV, INERIS, MIV/TEAM, 60550 Verneuil-en-Halatte, France; 3Department of Endocrinology, Diabetes Mellitus and Nutrition, Amiens University Hospital, 1 Rond Point du Pr Christian Cabrol, 80054 Amiens, France

**Keywords:** radiofrequency, brown adipose tissue, UCP1-dependent thermogenesis, brown adipogenesis, rat model

## Abstract

The widespread use of wireless technologies has raised public health concerns about the biological effects of radiofrequency (RF) exposure. Children have a higher specific absorption rate (SAR) of radiation energy compared to adults. Furthermore, brown adipose tissue (BAT) is more prevalent in infants and tends to decrease with age. Previous animal studies demonstrated a cold sensation in rats exposed to 900 MHz (second generation, 2G). UCP1-dependent thermogenesis and BAT hyperplasia are two fundamental adaptive mechanisms initiated in response to cold. This study investigated the impact of short-term exposure to 2G and fifth generation (5G) on key thermogenic and adipogenic markers related to these mechanisms while considering age and exposure duration. Juvenile and young adult Wistar rats were randomized into three subgroups: a 5G group (3.5 GHz), 2G group (900 MHz), and a control group (SHAM). They were exposed to their respective continuous-wave RF signals for 1 or 2 weeks at an intensity of 1.5 V/m, with two exposure sessions of 1 h per day. After the exposure period, a RT-qPCR was carried out to evaluate the genetic markers involved in BAT thermogenesis and adipogenesis. Two adipogenic biomarkers were affected; a fold change reduction of 49% and 32% was detected for *PRDM16* (*p* = 0.016) and *C/EBP β* (*p* = 0.0002), respectively, after 5G exposure, regardless of age and exposure duration. No significant RF effect was found on UCP1-dependent thermogenesis at a transcriptional level. These findings suggest that exposure to a 5G radiofrequency may partially disrupt brown adipocyte differentiation and thermogenic function by downregulating *PRDM16* and *C/EBP β*, possibly leading to higher cold sensitivity.

## 1. Introduction

The emergence of the fifth generation (5G) of wireless cellular technology has generated widespread interest and ongoing debate. Some frequency bands are conserved across mobile network generations, such as 900 MHz, which is still widely used in 2G and 3G networks. The 5G network is the latest advent in wireless communication, introducing the new 3.5 GHz frequency band. While this new technology offers major advances such as reduced latency, enhanced connectivity, and increased device density, the biological effects of low-level radiofrequency (RF) electromagnetic radiation on public health remain a major concern.

The International Commission on Non-Ionizing Radiation Protection (ICNIRP) has established guidelines and restrictions for exposure to radiofrequency electromagnetic fields from 100 kHz to 300 GHz to avoid potential health risks and ensure environmental safety. Regulatory limits for whole-body exposure to RF are set at 0.08 W/kg for the general public and 0.4 W/kg for occupational exposure to ensure safe working conditions. Despite assumptions that RF exposure below the regulatory limits should not cause temperature disruption, studies in animals [1,2,3,4,5,6] and humans [7,8] have shown the opposite. Vasoconstriction and a shift in thermal preference towards warmer temperatures were observed in rats exposed to 2G RF signals (900 MHz) at low intensity levels [3,5]. These physiological responses are similar to a cold sensation. However, the underlying molecular mechanisms behind these responses are unclear and the effects of 5G radiofrequency (RF) exposure on thermoregulation are unknown.

In the present study, we hypothesize that RF exposure may trigger the molecular mechanisms behind cold-induced responses in brown adipose tissue (BAT). In fact, BAT is the main effector of heat production in the body and is frequently stimulated during cold to maintain the internal body temperature in mammals [9,10]. Non-shivering thermogenesis and BAT hyperplasia are two fundamental adaptive thermoregulatory responses to cold stress [11]. Non-shivering thermogenesis is a mechanism by which BAT generates heat in mammals. This process is mediated primarily by the uncoupling protein UCP1 in the mitochondria. UCP1 dissipates the proton gradient to reduce ATP production and instead produce energy in the form of heat [12]. UCP1 activity is regulated by a complex network involving the PPAR signaling pathway, with key markers like *PGC1 α* (peroxisome proliferator-activated receptor gamma co-activator-1 alpha), *PPAR α* (peroxisome proliferator-activated receptor alpha), and *PPAR γ* (peroxisome proliferator-activated receptor gamma). These transcriptional factors play a crucial role in mitochondrial biogenesis and lipid metabolism, enhancing heat production. Cidea (cell death-inducing DNA fragmentation factor-like effector A) is also an interesting indicator of lipid metabolism [13,14], although its relationship to UCP1 remains unclear. All these factors are part of the BAT-specific gene transcriptional program that regulates the activity of UCP1 [15,16].

Brown adipose tissue hyperplasia is also crucial for adapting to cold-inducible environments. This mechanism is primarily driven by brown adipogenesis, which is characterized by the proliferation and differentiation of mature brown adipocytes from BAT precursors. This process is regulated by adipogenic markers such as C/EBP α (CCAAT enhancer binding protein alpha), C/EBP β (CCAAT enhancer binding protein beta), and PRDM16 (PR domain 16). PRDM16 is crucial for BAT development, playing a vital role in maintaining a brown phenotype identity and a functional thermogenic capacity [17,18]. Zfp423 (zing finger protein 423) is an indicator of preadipocyte commitment [19] and is also important for the development of adipose tissues. The plasticity of adipose tissue is well known and important for adaptation to different thermal conditions [20]. BAT thermogenesis is mediated by the β3-adrenergic signaling pathway. The sympathetic activation of BAT by noradrenaline triggers these thermogenic mechanisms in response to cold [21]. Furthermore, researchers have demonstrated the possible involvement of the calsyntenin 3 (Clstn3β)-S100b axis in the regulation of this adrenergic pathway, with S100b as its downstream effector [22].

In the present study, we investigated the effects of exposure to 900 MHz (2G) and 3.5 GHz (5G) on BAT thermogenic and adipogenic markers. Previously, an elevated body temperature was observed in mice during two one-hour daily sessions from the second to the seventh day of exposure to 900 MHz (2G) [4]. The authors suggested in a second study that this increase in temperature might be due to the activation of BAT thermogenesis in response to RF exposure [3]. In this study, rats were used to assess RF exposure effects, since they are an appropriate physiological model in which to investigate BAT thermogenesis and adipogenesis in mammals. We hypothesized a potential age-related difference in responses to RF exposure, as children have a higher specific absorption rate (SAR) of radiation energy compared to adults [23,24]. This may be attributed to physiological and anatomical differences during the human developmental period [25]. Furthermore, BAT is more abundant in children than in adults, indicating distinct physiological adaptations to environmental stress, particularly in terms of thermoregulation [26]. Hence, juvenile and young adult rats were both assessed.

In the present study, we used a similar RF exposure setup to that of Mai et al. [4] with exposure durations of 1 and 2 weeks to observe the potential adaptive response of thermogenesis. We applied an intensity level of around 1.5 V/m during each RF exposure session to mimic the current range of environmental RF exposure [27]. The specific effects of 900 MHz (2G) and 3.5 GHz (5G) exposures on BAT thermogenesis and adipogenesis are unknown. In this research, we used a transcriptional approach by RT-qPCR to investigate these mechanisms in juvenile and young adult rats. We hypothesized that cold-induced transcriptional changes [28,29] may occur in rats exposed to 2G and 5G RF signals, but these effects may differ depending on age and exposure duration. This study is the first to evaluate the specific effects of 900 MHz (2G) and 3.5 GHz (5G) radiofrequency exposures on these thermoregulatory mechanisms while considering the possible influence of age and exposure duration on these effects.

## 2. Results

### 2.1. UCP3 mRNA Expression Is Downregulated in 5G-Exposed Rats Compared to Those Exposed to 2G

The uncoupling proteins UCP1 or UCP3 play a vital role in regulating thermogenesis and the energy balance in response to cold. These mitochondrial transport proteins are activated in brown adipose tissues during cold sensation. In this study, we assessed the effects of 2G and 5G radiofrequency exposures on *UCP1* and *UCP3* mRNA levels in juvenile and young adult rats. An ANOVA revealed the significant effect of RF on *UCP3* mRNA expression (*p* = 0.0432). An indicative trend was noted for the age factor (*p* = 0.051), while the duration period showed no significant effect on *UCP3*’s relative gene expression. As no significant interactions were found among these factors (RF exposure, age, and duration period), data for both age groups were pooled per RF group independently of the duration period for a more robust analysis. These consolidated data were analyzed by a Kruskal–Wallis test, followed by two-tailed Mann–Whitney tests. The results showed a significant reduction in *UCP3* mRNA levels (*p* = 0.0305) in the 5G group compared to the 2G group (Figure 1A). However, no significant effect was observed in either the 2G group or in the 5G group in comparison to the control group. As for *UCP1*, no significant effect or interaction was found among the groups (Figure 1B).

### 2.2. Fifth-Generation Radiofrequency Exposure Reduced the mRNA Levels of Adipogenic C/EBP β, Zfp423, and PRDM16 Markers in BAT

BAT hyperplasia is a physiological phenomenon commonly observed during cold exposure. This adaptive response is primarily driven by the differentiation and maturation of brown adipocytes from BAT precursors. A specific rise in *C/EBP α*, *C/EBP β*, and *PRDM16* mRNA levels is often detected during this process. *Zfp423* is an important transcriptional regulator of preadipocyte commitment and development. These transcription factors are important for BAT development. In this study, we also investigated the effects of RF exposure on these key brown adipogenic markers. An ANOVA revealed a significant effect of RF exposure on *C/EBP β* (*p* = 0.0008) and *Zfp423* (*p* = 0.0137), while an indicative effect was detected in *C/EBP α* (*p* = 0.069) and *PRDM16* (*p* = 0.0893) mRNA levels. No significant effects of age or exposure duration and no interactions among these different factors were found for these markers. Data were consolidated using the same process as described above and were analyzed by Kruskal–Wallis and two-tailed Mann–Whitney tests. Decreased mRNA levels of *C/EBP α* (*p* = 0.0431), *C/EBP β* (*p* = 0.0014), and *Zfp423* (*p* = 0.007) were observed in the 5G group compared to those in the 2G group (Figure 2A–D). Interestingly, a marked reduction was observed for *PRDM16* (*p* = 0.016) and the other adipogenic markers (*C/EBP β*, *p* = 0.0002; *Zfp423*, *p* = 0.009) in the 5G group compared to the control group (Figure 2A–D), except for *C/EBP α*, whose reduction did not reach statistical significance (*p* = 0.11). The 5G group exhibited a fold change of approximately 0.51 for *C/EBP β* and 0.68 for *PRDM16* relative to the control group, indicating a decrease of 49% and 32%, respectively. For *Zfp423*, a fold change of 0.70 was observed in the 5G group compared to the control group, reflecting a significant reduction of approximately 30%. No significant changes were found between the control and 2G-exposed rats in terms of these markers.

### 2.3. Age-Dependent Modulation of PPAR α mRNA Expression After 2G Exposure

The PPAR transcriptional cascade plays a vital role in regulating BAT thermogenesis. We studied the effects of RF exposure on *PPAR α*, *PGC1 α*, *PPAR γ*, and *Cidea* in juvenile and young adult rats at the transcriptional level. No significant RF effect or interaction was detected for *PGC1 α*, *PPAR γ*, or *Cidea* (Figure 3A–C). However, a significant interaction between RF exposure and age was found for *PPAR α* (*p* = 0.015). To further clarify this interaction, data from the 1-week and 2-week exposure periods were regrouped by RF exposure condition for juvenile and young adult rats, respectively. Then, we applied the Kruskal–Wallis test followed by the two-tailed Mann–Whitney test for comparisons between experimental groups. After 2G exposure, a marked reduction in *PPAR α* (*p* = 0.0025) was found in young adult rats in comparison to juvenile rats (Figure 3D). However, no such effect was detected after 5G exposure. Furthermore, in young adult rats, *PPAR α* mRNA levels were significantly decreased in the 2G group compared to the control group (*p* = 0.0494), with a fold change of 0.54 seen, indicating a reduction of approximately 46%. Moreover, the 5G-exposed group of young adult rats also showed a trend toward decreasing (*p* = 0.0696) *PPAR α* mRNA levels compared to those in the control group. No significant changes were found between non-exposed and RF-exposed juvenile rats for this marker. These findings suggest that 2G exposure could alter *PPAR α* mRNA expression depending on age.

### 2.4. ADRβ3 mRNA Expression Was Modulated by Age, Exposure Duration, and RF Exposure

An elevated innervation of BAT is triggered during the thermoregulatory response to cold. This innervation is mediated by the pre-thermogenic receptor of adrenergic signaling (ADRβ3) and is regulated by the Clstn3β-S100B pathway, with S100B as its downstream effector. Here, we assessed the effects of RF exposure and its possible interactions with age or the exposure duration on *ADRβ3* and *S100B* mRNA levels. Our findings revealed a significant interaction between RF, age, and exposure duration for *ADRβ3* (*p* = 0.0378). After a 1-week exposure to RF, a significant RF effect was observed in juvenile rats (*p* = 0.038), with higher *ADRβ3* mRNA levels in the 5G and 2G groups in comparison to the control group (*p* = 0.0472). After a 2-week exposure to RF, these differences were not detected between the different groups of juvenile rats.

Interestingly, an age-related RF effect on *ADRβ3* mRNA expression was only observed after a 1-week exposure to RF. As shown in Figure 4A, *ADRβ3* mRNA levels were significantly higher in juvenile rats compared to young adult rats after a 1-week exposure to the 5G RF signal (*p* = 0.009). After a 1-week exposure to 2G, a similar trend was detected between the two age groups, although it did not reach statistical significance (*p* = 0.0758). No age-related RF effect was found after the 2-week exposure.

We also noted that the exposure duration had an effect on this adrenergic marker. *ADRβ3* levels in juvenile rats were significantly lower after a 2-week exposure to 5G compared to a 1-week exposure (*p* = 0.0143). A similar trend was observed in 2G-exposed juvenile rats, except that it did not reach statistical significance (*p* = 0.0758). No exposure duration effect was detected when comparing unexposed and RF-exposed young adult rats. As for *S100B* mRNA expression, no significant RF effect or interaction was observed (Figure 4B).

## 3. Discussion

In the present study, we investigated the effects of short-term 5G (3.5 GHz) and 2G (900 MHz) exposures on non-shivering thermogenesis and brown adipogenesis in juvenile and young adult rats. In prior studies, vasoconstriction and an increase in circulating levels of noradrenaline and non-esterified fatty acids were observed in rats exposed to 900 MHz RF signals at a low intensity [4,6]. These cold-inducible responses are often followed by the activation of UCP1-dependent thermogenesis and BAT hyperplasia to maintain thermal balance [11]. In this study, we focused on the transcriptional mechanisms of BAT thermogenesis and adipogenesis to understand RF-induced thermoregulatory changes.

### 3.1. The Core Transcriptional Pathway of UCP1 Thermogenesis Is Not Impacted by RF Exposure

BAT thermogenesis is mediated by the uncoupling protein UCP1 and key factors involved in the PPAR signaling pathway: *PGC1 α*, *PPAR α*, *PPAR γ*, *Cidea*, and *PRDM16*. *PGC1 α* is a vital regulator of cold-induced BAT thermogenesis. It is a transcriptional coactivator that promotes *UCP1* expression through a PPAR transcriptional cascade. It enhances key biological functions essential for heat production through the activation of nuclear receptors and transcription factors such as *PPAR γ* and *PPAR α* [30,31]. Its interaction with the nuclear receptor peroxisome proliferator activated receptor gamma (*PPAR γ*) facilitates mitochondrial biogenesis in BAT [32]. *PGC1 α* binds coordinately with *PPAR α* to increase lipid catabolism for BAT thermogenesis [33]. Cidea was later discovered to be a lipid droplet-associated protein [13,14,34] which is highly present in brown adipocytes [35,36]. However, its relationship to UCP1 remains unclear. Some studies have demonstrated that Cidea reduced UCP1 activity in BAT [37]. Other studies showed that its expression is positively correlated with UCP1 activity in brown and beige adipocytes [38,39]. Thus, Cidea is universally considered to be a brown adipocyte marker, involved in regulating the lipid content in BAT. All these biomarkers are involved in the core transcriptional pathway of UCP1-dependent thermogenesis [15,16].

Our study showed no significant effects of RF exposure on *UCP1*, *PGC1 α*, *PPAR γ*, and *Cidea*, regardless of age and exposure duration. Maalouf et al. [40] also found no significant changes in *UCP1*’s relative mRNA expression in mice exposed to 2G (900 MHz) for one week. Concerning *UCP3* mRNA levels, a significant difference was only observed between the 2G- and 5G-exposed groups, and not when these groups were compared to the controls. Some researchers suggest that UCP3 is also a mediator of thermogenesis [41,42] and contributes to energy homeostasis [43]. Several studies have reported transcriptional changes in UCP1-dependent thermogenesis in BAT after cold exposure [28,29]. These changes were not observed in our study, suggesting that the core transcriptional pathway of UCP1-dependent thermogenesis is not impacted by short-term RF exposure. We hypothesize that maybe a different thermogenic pathway, independent of UCP1, might be impacted by RF exposure to maintain energy homeostasis. In fact, it has recently been accepted that non-shivering thermogenesis does not solely rely on UCP1 activity [44,45]. The second hypothesis is that *UCP1* transcriptional changes are observed only after the first few days of RF exposure, and not after prolonged exposures of 7 or 14 days. In fact, it has been reported that *UCP1* mRNA expression increases rapidly during the initial phase of acute cold exposure to enhance the body’s heat production and thermogenic capacity. However, after prolonged cold exposure, *UCP1* mRNA levels are significantly lower compared to the elevated levels observed during the acute phase of cold exposure [46]. Thus, long-term adaptation to the cold does not necessarily require UCP1 [47], as the enhanced thermogenic capacity acquired during the early phase of cold exposure is maintained, ensuring efficient heat production. Previous studies found that rats exposed to RF maintained vasoconstriction in their tails, the primary organ for heat exchange, to reduce heat loss [3,5]. This phenomenon is sufficient to maintain a constant body temperature within the thermoneutral zone without the activation of thermogenesis [48]. In our case, to confirm this hypothesis, it would be interesting to measure peripheral tail temperatures in RF-exposed rats.

### 3.2. Does Short-Term 5G Exposure Reduce Brown Adipogenesis?

BAT hyperplasia is an adaptive physiological response to cold exposure driven by the persistent differentiation and recruitment of thermogenic brown adipocytes [49,50]. PRDM16 is known to play an important role in BAT thermogenesis and brown adipogenesis [51]. In our research, we also investigated biomarkers of the C/EBP family, which work with PRDM16 in the transcriptional regulation of brown adipocyte differentiation [52]. Studies have highlighted the importance of *C/EBP β* and *PRDM16* in the differentiation of brown adipocytes [53]. *C/EBP β* and *PRDM16* form a transcriptional complex and initiate the early phase of brown adipogenesis [53,54,55]. *PRDM16* is the main regulator of this process and is required for the maintenance of the thermogenic function and phenotype of brown adipocytes [17,18,51]. Another marker, *C/EBP α*, plays a crucial role in the terminal differentiation phase of adipogenesis [56]. The knock-down of *C/EBP α* impaired the ability to store and accumulate lipid droplets in brown adipose tissue, disrupting both adipogenesis and the lipid metabolism essential for heat production [57]. *C/EBP α*, *C/EBP β*, and *PRDM16* are vital regulators for obtaining fully matured and differentiated brown adipocytes [52]. Gupta et al. [58] has identified *Zfp423* as a transcriptional regulator of preadipocyte determination. Furthermore, they demonstrated an impaired development of both brown and white adipose tissues in Zfp423 knockout mice [19]. Although its role in the thermogenic capacity of brown adipocytes remains unclear, it may promote the lineage commitment and development of adipocytes [59].

In our study, we observed a fold decrease in *PRDM16* (49%), *C/EBP β* (32%), and *Zfp423* (30%) after 5G exposure, regardless of age and exposure duration. This downregulation after RF exposure could potentially explain the increased cold sensitivity reported in several studies [3,5]. For example, (i) the low levels of *PRDM16* and *C/EBP β* could partially impair the thermogenic function and differentiation of BAT, which may contribute to a reduced recruitment of thermogenic brown adipocytes in 5G-exposed rats while (ii) the decreased levels of *Zfp423* might hinder the lineage commitment and development of adipocytes. The possibility that brown adipogenesis may be reduced after 5G exposure might result in fewer thermogenic brown adipocytes being available. This modification could contribute to a decrease in overall heat production by BAT, leading to the emergence of a cold thermal sensation after RF exposure.

Surprisingly, these transcriptional responses were not observed after the 2G exposure. One possible explanation could be the differences in the physical properties of these RF signals, which may lead to distinct biological adaptations and stress responses, as evidenced in other studies [60]. The characteristics of RF waves, such as their frequency, may influence the pattern of their energy deposition, leading to a selective absorption of radiation energy in specific tissue beds [61]. The superficial localization of interscapular brown adipose tissue perhaps makes it more sensitive to the biological effects of radiofrequency electromagnetic waves emitted at 3.5 GHz rather than at 900 MHz.

### 3.3. RF Exposure Amplifies Age-Related PPAR α Downregulation

Few studies have explored how the potential effects of RF are associated with age, especially in terms of thermoregulation. In fact, it is widely known that BAT is more prevalent in infants and tends to decrease with age [62]. However, this does not necessarily reflect reduced BAT activity in older individuals [63]. The originality of our work lies in the investigation of a possible interaction between the effects of RF exposure and age on thermoregulation. Among all the studied markers, an interaction was found only for *PPAR α* after 2G exposure, with a fold decrease in young adult rats compared to juvenile rats. Interestingly, within the young adult group, RF exposures (both 2G and 5G) led to a lower genetic expression of *PPAR α* compared to the controls. Our study is the first to demonstrate that RF exposure may exacerbate the potentially age-dependent downregulation of this lipid metabolism regulator.

### 3.4. Age-Related Effect of RF on ADRβ3 After 1-Week Exposure

The role of β3-Adrenergic Receptor (ADRβ3) in initiating BAT thermogenesis has been well documented in rodents exposed to cold environments [64]. ADRβ3 activation in BAT induces lipolysis, leading to the release of non-esterified fatty acids (NEFAs). NEFAs are simultaneously used as substrates and in turn activate mitochondrial UCP1 for heat production. Other studies have shown that cold exposure also triggers the proliferation of mature brown adipocytes through ADRβ3 activation [21]. Our data reported an age-related effect of RF on *ADRβ3* mRNA expression after a 1-week exposure to RF signals. *ADRβ3* mRNA levels seem to be reduced in older rats compared to juveniles after a 1-week exposure to 5G or 2G RF signals. No similar effects were observed between the different age groups after a 2-week exposure. We also observed an effect of the exposure duration on the RF-exposed juvenile rats. A lower *ADRβ3* expression was detected after a 2-week exposure to the 5G RF signal compared to a 1-week exposure. A similar trend was noted between 2G-exposed juvenile rats. Regarding the main effects of these RF signals, 2G-exposed and 5G-exposed juvenile rats showed higher *ADRβ3* mRNA levels compared to controls, but only after a 1-week exposure. These results suggest a potential effect of RF that is associated with age and exposure duration. It is important to note that this interaction was observed exclusively in *ADRβ3* among all the tested markers. Concerning *S100b*, our study found no significant findings after any RF exposures. According to the literature, this downstream effector of the Clstn3β-S100B axis is induced in BAT after cold exposure [22] and may regulate the β3-adrenergic pathway.

### 3.5. Limitations of This Study

To the best of our knowledge, this study is the first to evaluate the effects of 5G and 2G radiofrequency exposures on BAT thermogenesis and adipogenesis while considering age and exposure duration factors. However, we have encountered several limitations in our research. In this work, we have carried out a relative gene expression study by using RT-qPCR to evaluate the biomarkers involved in non-shivering thermogenesis and brown adipogenesis. It is important to corroborate these findings with a proteomic analysis in order to confirm the observed biological effects. In fact, researchers have shown that the mRNA levels of thermogenic markers such as *UCP1* and *PGC1 α* do not necessarily reflect their corresponding protein levels [65,66]. In this study, we could not obtain sufficient BAT samples during organ collection, especially from older rats, for proteomic analysis. Another limitation to our setup was having only five rats per experimental condition, since one climatic chamber could accommodate a maximum of five rat cages. It is possible that some biomarkers did not reach statistical significance due to the limited sample size.

## 4. Materials and Methods

### 4.1. Ethics Statement

All experimental procedures were approved by the Regional Ethical Committee for Health, Animal and Environment Protection (Amiens, France) and the French Ministry of Research (APAFIS 34412#) in accordance with the European guidelines (2010/63/EU) and the French governmental decree 2013-118 on laboratory animal care.

### 4.2. Animal Housing

Sixty male Wistar rats were purchased from Janvier Labs (Le Genest Saint Isle, France). They were individually housed in plexiglass cages placed in one of the two climatic chambers and kept under controlled air temperature (24.2 ± 0.3 °C) and humidity conditions (42.6 ± 6.6%) in a 12:12 h light/dark cycle. One chamber was used to expose animals to RF field signals and the other one was used for house animals not exposed to RFs. The animals were acclimatized for 4 days, with free access to food chow (3436EXF12, Serlab, Montataire, France) and drinking water. Animal care was carried out on a daily basis between 5 and 6 p.m.

### 4.3. Experimental Design

Thirty juvenile rats (3-week-old) and thirty young adult rats (8-week-old) were randomly assigned to 12 groups. Six groups were composed of juvenile rats and the other six groups were composed of young adult rats, with 5 animals per group (n = 5). For each age group there were 6 groups. Three of them were control, 5G-exposed and 2G-exposed groups for the one-week exposure duration, while the other three groups subjected to each type of exposure were kept under the same experimental conditions and exposed to their respective RF signals for a duration period of two weeks (Figure 5A).

### 4.4. RF Exposure System

The 5G group was exposed to a continuous-wave RF signal set at 3.5 GHz and the 2G group to one at 900 MHz, with an exposure period of one or two weeks. The two one-hour RF exposure sessions per day were applied at an intensity of 1.5 V/m. These sessions were scheduled randomly; one in the morning and one in the afternoon. According to the French national frequency agency (ANFR), this intensity level reflects our current environmental exposure when using wireless network technologies and mobile phones [27].

A generator (AnaPico ASPIN4010—9 kHz–4000 MHz, Glattbrugg, Switzerland), located outside the climatic chambers, was set to produce a 3.5 GHz band signal for 5G exposure. It was coupled with an amplifier RFPA (RF26003800-4x0.5W) connected to one antenna (antenna Laird multi-band CFS60383) inside the chambers. For 2G exposure, the same generator was set to 900 MHz, paired with another amplifier RFPA (RFS7002500-6x0.5) capable of emitting this RF band, and connected to two antennas (Kathrein 800-10465, Rosenheim, Germany).

The antennas were aligned horizontally in the climatic chamber, 80 cm above the exposed rats’ boxes, thus at a height larger than 2.4λ and 9λ, at 900 MHz and 3.5 GHz, respectively, with λ representing the wavelength. The position of the antennas was adjusted to minimize the variation in the field amplitude within each cage and between cages.

Using an electric field probe EP600 (Narda Safety Test Solutions, Cisano sul Neva, Italy), we measured the intensity level in five different positions in each cage. An electric field of 1.6 ± 0.4 V/m was measured under 5G exposure and 1.6 ± 0.5 V/m under 2G exposure. These data were recorded on WinEP600 (Narda Safety Test Solutions, Cisano sul Neva, Italy). The transmitting device did not generate a static magnetic field. For the non-exposed groups (controls), the antennas in the adjacent climatic chamber remained unconnected to the generator.

Using these data, the mean intensity of the RF signal per cage was used to estimate the mean whole-body SAR during the experiment. The mean whole-body SAR was calculated to be 0.07 mW/kg for the 5G exposure and 0.24 mW/kg for the 2G exposure, following the method described by Mai et al. [4].

### 4.5. Organ Collection

After the RF exposure period, rats were euthanized by cardiac puncture under anesthesia using a mixture of air and 2.5% isoflurane (Iso-Vet 1000 mg/g, Piramal Healthcare UK Ltd., Morpeth, UK). Brown adipose tissues were collected from the interscapular region and preserved overnight in RNA later solution (Invitrogen, Carlsbad, CA, USA) at 4 °C and then stored indefinitely at −80 °C before use.

### 4.6. Relative Gene Expression: RT-qPCR

The BAT weight was measured and no significant difference was observed between unexposed and RF-exposed rats in both age groups (Appendix A).

Total RNA was isolated from brown adipose tissues using Qiazol (Qiagen, Germany) and then purified with RNA mini-spin columns. RNA extraction was carried out using the RNeasy Lipid Tissue Mini Kit (Qiagen, Hilden, Germany) according to the manufacturer’s instructions. RNA concentration and purity were assessed using a NeoDot spectrophotometer (Neo Biotech, Nanterre, France). Only good-quality RNA samples with a 260/280 ratio between 1.8 and 2.0 were selected for reverse transcription.

From each sample, 2 µg of RNA was reverse-transcribed using the High-Capacity cDNA Reverse Transcription kit with RNase inhibitor (Applied Biosystems, Foster City, MA, USA). The reverse transcription was performed on a thermal cycler VeritiPro (Applied Biosystems, Foster City, MA, USA) using the programmed conditions recommended by the manufacturer. The synthesized cDNA samples were diluted to 10 ng/µL with nuclease-free water DEPC (Invitrogen, Carlsbad, CA, USA) before storage at −20 °C until use for real-time PCR.

Key biomarkers involved in non-shivering thermogenesis and BAT adipogenesis were analyzed by RT-qPCR (Figure 5B): *UCP1*, *UCP3*, *PGC1 α*, *PPAR α*, *PPAR γ*, *Cidea*, *ADRβ3*, *S100B*, *C/EBP α*, *C/EBP β*, *ZFP423*, and *PRDM16*. A total of 10 ng of cDNA and a final concentration of 0.4 µM of each specific primer were mixed with SYBR green master mix (Applied biosystems, Foster City, MA, USA). All reactions were performed in duplicate on a 96-well PCR plate (Starlab, Barcelona, Spain) and quantitative RT-qPCR was carried out on an ABI 7900 system (Applied Biosystems, Foster City, MA, USA).

Relative gene expression was calculated using the 2^−ΔΔCT^ method. All data were normalized to the gene *TBP*. *TBP* was used as endogenous control as it showed the most stable mRNA expression among all tested housekeeping genes (*36B4*, *Ppib*, *TFIIB*), exhibiting the lowest coefficient of variation across all samples.

All primers (Invitrogen, Carlsbad, CA, USA) used for qPCR experiments are listed in Table 1.

### 4.7. Statistical Analysis

Data were presented as mean ± SEM. Statistical analysis was performed on Statview software version 5.0 and graphical illustrations were generated using GraphPad Prism 9. The data’s normality was verified with Shapiro–Wilk tests. Relative gene expression data were analyzed by a three-way analysis of variance (ANOVA) with the frequency of exposure, exposure duration, and age as subject factors. The gene expression data for each tested biomarker were nonparametric, with n = 5 per experimental condition for both age groups (juvenile and young adult). A Kruskal–Wallis test followed by a Mann–Whitney test were performed for multiple comparisons between groups when the ANOVA revealed a significant effect from one factor or an interaction between two or three factors. The threshold for statistical significance was set at * *p* < 0.05, while a difference approaching significance was indicated at ^i^ *p* < 0.1 for all analyses.

## 5. Conclusions

Our research showed a fold change decrease of 49% for *PRDM16* and 32% for *C/EBP β* in terms of their mRNA levels after exposure to 5G. As mentioned previously, these adipogenic markers are important in the differentiation and maturation of brown adipocytes from BAT precursors, as well as the maintenance of their thermogenic capacity. In contrast, UCP1-dependent thermogenesis was not impacted by RF exposure at the transcriptional level. Most of the studied thermogenic markers showed no age-related or exposure duration-related effects associated with RF exposure, except for *PPAR α* and *ADRβ3*. This study provides new insights into the potential impact of 5G exposure on brown adipogenesis. The disrupted differentiation and thermogenic capacity of brown adipocytes through *PRDM16* and *C/EBP β* downregulation may affect the development and characteristics of BAT, potentially leading to increased cold sensitivity after RF exposure. These findings could partially explain the physiological events related to cold stress seen after RF exposures. However, it is important to investigate the peripheral tail temperature in rats in order to confirm our hypothesis and explain the observed biological effects. Furthermore, it may provide valuable insights to help us better understand the impact of low-intensity 2G and 5G RF signals on vasomotor responses. Most studies on low-level radiofrequency exposure have predominantly focused on “non-thermal” biological effects, such as oxidative stress, genetic instability, and reproductive health, although these findings remain heterogeneous [77]. The impact of low-intensity RFs on thermoregulation remains largely an uncharted aspect of environmental health and safety. This research addresses this gap and can help to raise public awareness about the potential health risks posed by radiofrequency electromagnetic radiation, particularly 5G, with the rise in wireless technologies.

## Figures and Tables

**Figure 1 ijms-26-02792-f001:**
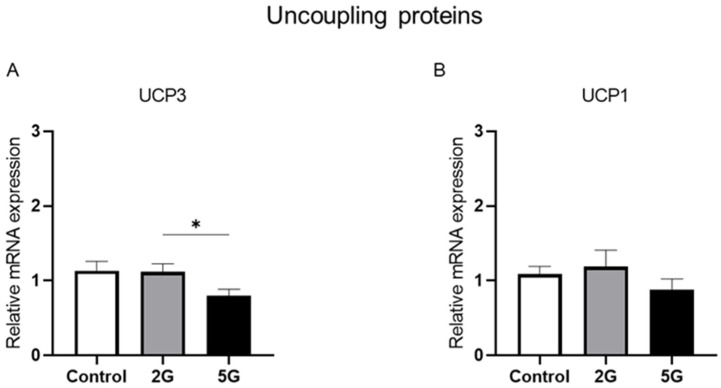
*UCP3* mRNA levels in BAT differed between RF-exposed groups. Juvenile and young adult rats were either not exposed (control) or exposed to 2G or 5G RF signals for 1 or 2 weeks. Brown adipose tissues were collected and processed for RT-qPCR analysis of mRNA expression of uncoupling proteins (**A**) *UCP3* and (**B**) *UCP1*. Data for both age groups and duration periods were pooled per RF exposure condition. *n* = 19 for control group and *n* = 19–20 for each RF group. Values were presented as mean ± SEM and analyzed by non-parametric Kruskal–Wallis tests followed by two-tailed Mann–Whitney tests. Significance * *p* < 0.05 based on Mann–Whitney test.

**Figure 2 ijms-26-02792-f002:**
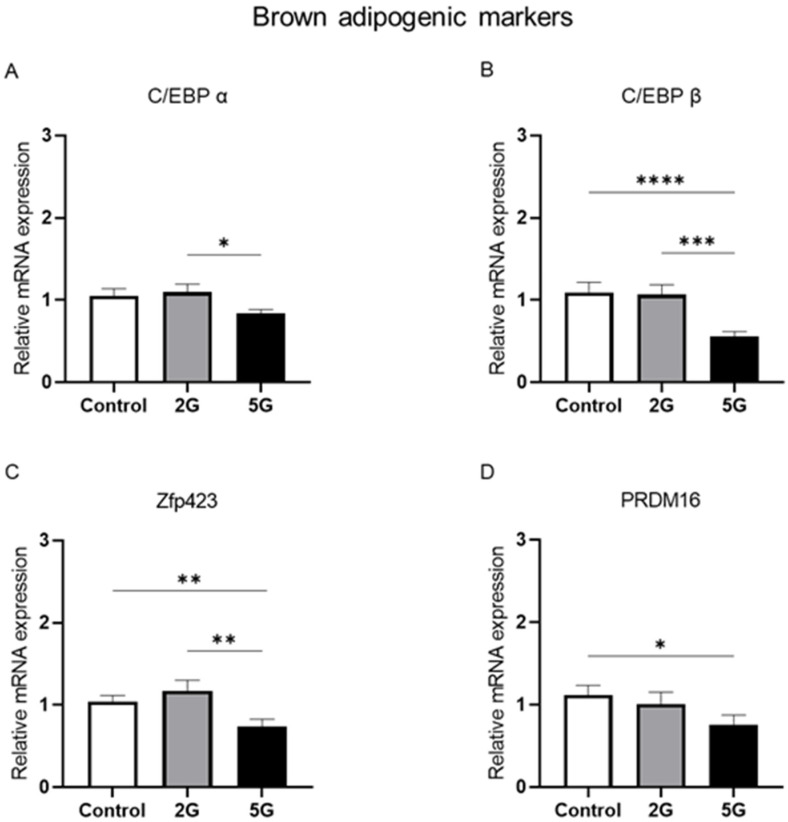
*C/EBP β*, *Zfp423*, and *PRDM16* mRNA expressions in BAT are downregulated after 5G exposure. Juvenile and young adult rats were not exposed (control) or exposed to 2G or 5G RF signals for 1 or 2 weeks. Brown adipose tissues were collected and processed for RT-qPCR analysis of brown adipogenic markers: (**A**) *C/EBP α*, (**B**) *C/EBP β*, (**C**) *Zfp423*, and (**D**) *PRDM16*. Data for both age groups and duration periods were pooled per RF exposure condition. *n* = 19 for control group and *n* = 19–20 for each RF group. Values were presented as mean ± SEM and analyzed by non-parametric Kruskal–Wallis tests followed by two-tailed Mann–Whitney tests. Statistical significance was set at * *p* < 0.05, ** *p* < 0.01, *** *p* < 0.001, **** *p* < 0.0001, using Mann–Whitney test results.

**Figure 3 ijms-26-02792-f003:**
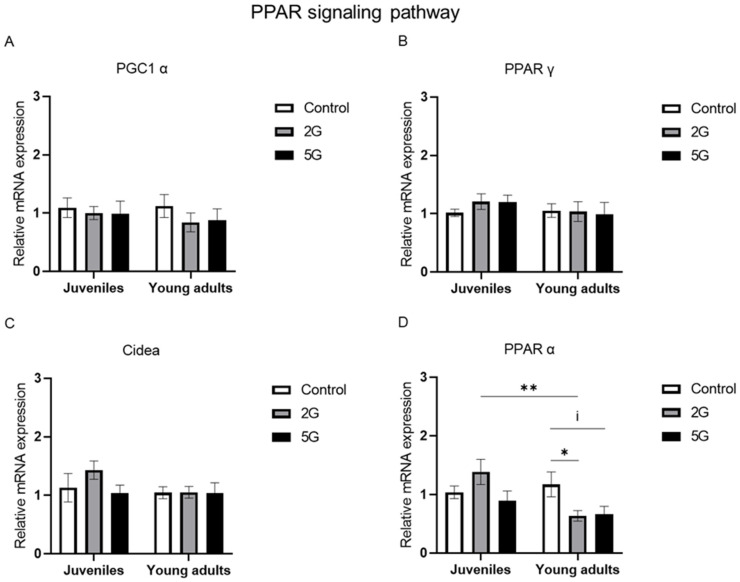
*PPAR α* mRNA levels are lower in young adult rats after 2G exposure. Juvenile and young adult rats were not exposed (control) or exposed to 2G or 5G RF signals for 1 or 2 weeks. Brown adipose tissues were collected and processed for RT-qPCR analysis of key regulators involved in the PPAR signaling pathway: (**A**) *PGC1 α*, (**B**) *PPAR γ*, (**C**) *Cidea*, and (**D**) *PPAR α*. Data from both exposure periods were regrouped per RF exposure condition for juvenile and young adult rats, with *n* = 9–10 per experimental condition for each age group. Values were presented as mean ± SEM and analyzed by two-way ANOVA. Non-parametric Kruskal–Wallis and two-tailed Mann–Whitney tests were conducted of *PPAR α*’s relative mRNA expression. Statistical significance was set at * *p* < 0.05 and ** *p* < 0.01 using Mann–Whitney test results. An indicative trend was noted at ^i^ *p* < 0.1.

**Figure 4 ijms-26-02792-f004:**
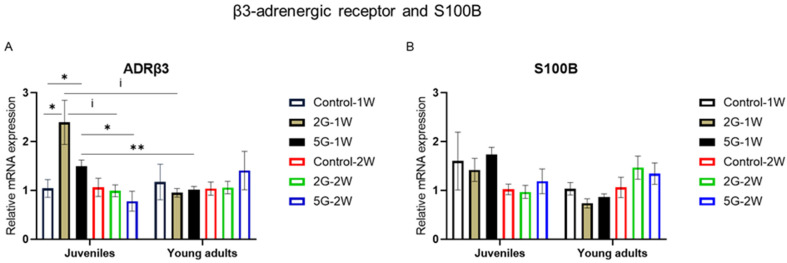
RF-mediated effects on β3-adrenergic receptor and *S100B* mRNA expression in juvenile and young adult rats exposed to RF signals for 1–2 weeks. Juvenile and young adult rats were not exposed (control) or exposed to 2G or 5G RF signals for 1 or 2 weeks (W), with *n* = 4–5 per experimental condition for each age group. Brown adipose tissues were collected and processed for RT-qPCR analysis of β3-adrenergic receptor (*ADRβ3*) (**A**) and *S100B* (**B**). Values were presented as mean ± SEM and analyzed by three-way ANOVA. Non-parametric Kruskal–Wallis and two-tailed Mann–Whitney tests were conducted on *ADRβ3*’s mRNA expression due to significant interactions between RF, age, and duration period. Statistical significance was defined as follows: * *p* < 0.05 and ** *p* < 0.01, based on Mann–Whitney test results. An indicative trend was noted at ^i^
*p* < 0.1.

**Figure 5 ijms-26-02792-f005:**
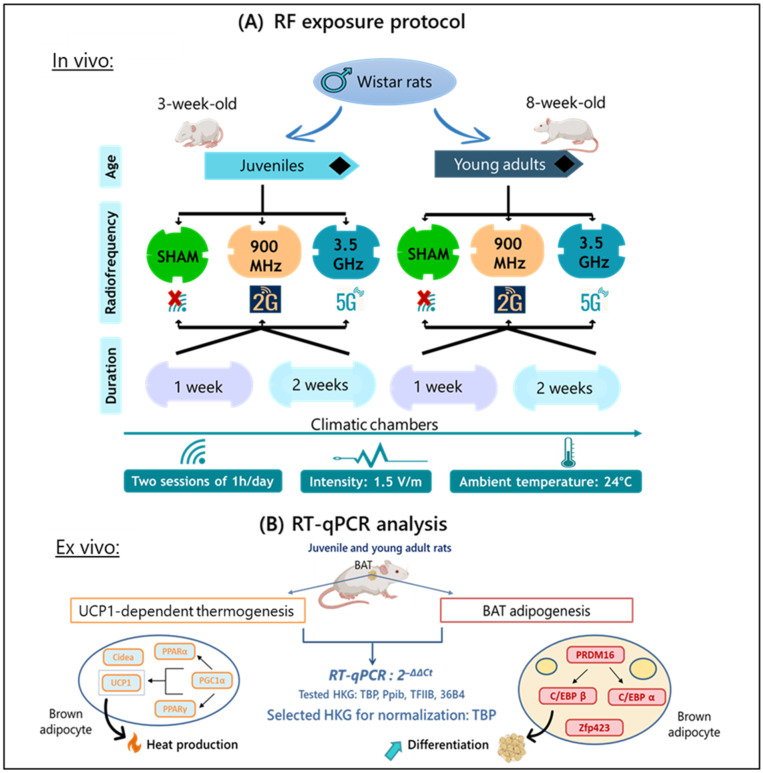
Study design: (**A**) RF exposure protocol and (**B**) RT-qPCR analysis of biomarkers related to UCP1-dependent thermogenesis and brown adipogenesis in interscapular BAT.

**Table 1 ijms-26-02792-t001:** List of qPCR primers and their specific gene targets. These primers were either derived from published scientific articles or designed using NCBI Primer-BLAST.

Function	Gene	Sequences (5′ to 3′)	Accession Number	Sequence Source
Housekeeping genes	*TBP*	F: CACCGTGAATCTTGGCTGTAAAC	NM_001004198.1	[67]
		R: CGCAGTTGTTCGTGGCTCTC		
	*TFIIB*	F: CCTGGCAGGAGTCCTATCTCT	NM_031041.2	[68]
		R: ACCAGCAATATCCCCGATTT		
	*Ppib*	F: CAGTCAAGACCTCCTGGCTAGA	NM_022536.2	Designed
		R: CCGTCTTGGTGTTCTCCACCTT		
	*36B4*	F: GGAACGTGGGCTTTGTGTTC	NM_022402.2	[69]
		R: GTACTGTGACCTCACACGGG		
Uncoupling proteins	*UCP1*	F: GCCTCTACGATACGGTCCAA	NM_012682.2	[70]
		R: TGCATTCTGACCTTCACCAC		
	*UCP3*	F: GACTCACAGGCAGCAAAGGAA	NM_013167.2	[71]
		R: GAGGAGATCAGCAAAACAGGC		
PPAR signaling pathway	*PPAR α*	F: ACTCGCAGGAAAGACTAGCA	NM_013196.2	[72]
		R: AGCAGTGGAAGAATCGGACC		
	*PGC1 α*	F: ATGGATATACTTTACGCAGGTCG	NM_031347.1	[73]
		R: TGGAAGCAGGGTCAAAATCG		
	*PPAR γ*	F: GTTGACACAGAGATGCCATTC	NM_013124.3	[74]
		R: CGCACTTTGGTATTCTTGGAG		
Brown adipogenic markers	*C/EBP α*	F: CGCTGTTGCTGAAGGAACTTGA	NM_001287577.1	Designed
		R: TTAGCATAGACGCGCACACTGA		
	*C/EBP β*	F: ACTTGATGCAATCCGGATCAAACG	NM_001301715.1	Designed
		R: CAGTTACACGTGTGTTGCGTCAGT		
	*Zfp423*	F: GCCAGATGACCTTCGAGAACGA	NM_001393718.1	Designed
		R: CGAACATCTGGTTGCACAGCTT		
	*PRDM16*	F: ACTTCGAGCTGCGAGAGTCC	NM_001427303.2	[75]
		R: GCAGCTCTCCTGGGATGACA		
β3-Adrenergic Signaling	*ADRβ3*	F: CTCACCGCTCAACAGGTTTGAT	NM_013108.2	Designed
		R: TTCTCCAGAAGTCAGGCTCCTT		
CLSTN3β pathway	*S100B*	F: TCAACAACGAGCTCTCTCACTT	NM_013191.2	Designed
		R: AGGCCATAAACTCCTGGAAGTC		
Thermogenic marker	*Cidea*	F: AGAAATGGACACCGGGCAAT	NM_001170467.1	[76]
		R: TGAAGCTTGTGCAGCGGATA		

## Data Availability

Data are available on request, via email, to the corresponding author.

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
