# Peer review of "5G Radiofrequency Exposure Reduces PRDM16 and C/EBP β mRNA Expression, Two Key Biomarkers for Brown Adipogenesis"

_ijms, 2025, doi:10.3390/ijms26062792_

Round 1
Reviewer 1 Report
Comments and Suggestions for Authors
The manuscript presents an original theme of great relevance for understanding the biological effects of non-ionizing radio frequencies. The authors have submitted a manuscript with concise and objective writing, facilitating the comprehension of the presented information. The study was very well designed, although the number of statistical repetitions is considered low. However, the authors discuss the study's limitations in a very honest manner. The results are presented in an organized and clear way. The discussion of the results is coherent and does not exceed the limits of the inferences allowed by the adopted methodology.
For these reasons, I consider the article suitable for publication. I only suggest minor corrections: the graphs use very similar color tones. I recommend that the authors hatch the columns for each group in the graph to facilitate the distinction between treatments. I also suggest that the authors provide the weight (with standard deviation) of the fat samples collected from the rats. This information is important for estimating the average fat volume of the rats, which could influence the results.
Finally, the authors should specify which endogenous gene was used as a control and whether the analysis method was delta CT or not. These details should be clearly stated in the text. Once these minor corrections are made, I believe the manuscript will be ready for publication.
Author Response
Comments 1: "I only suggest minor corrections: the graphs use very similar color tones. I recommend that the authors hatch the columns for each group in the graph to facilitate the distinction between treatments."
Response 1: Thank you for this helpful suggestion and we agree, especially for the figure 4, that it could be difficult to distinguish the different groups. For this reason, we have changed the color tones in the revised manuscript for the figures 1, 2 and 3 (white for control group, grey for 2G group and black for 5G group). For the figure 4, we kept these color tones for the one-week groups. As for the two-week groups, we have changed the color borders.
Comments 2: "I also suggest that the authors provide the weight (with standard deviation) of the fat samples collected from the rats. This information is important for estimating the average fat volume of the rats, which could influence the results."
Response 2: Thank you for your helpful suggestion. It is true that the fat weight could influence the results. Therefore, we have added supplementary data in an Excel file showing the average weight (± standard deviation) of the BAT samples collected from the juvenile and young adult rats respectively. The three-Way ANOVA analysis revealed no significant effect or interaction, highlighting that the weight of BAT did not differ significantly between non-exposed and RF-exposed rats in both juvenile and young adult groups. In the revised manuscript, we have also added one sentence in the beginning of the paragraph "relative gene expression: RT-qPCR" page 12, highlight in grey: "BAT weight was measured and no significant difference was observed between non-exposed and RF-exposed rats for both age groups (Supplementary Material S1, S2)."
Comments 3: "Finally, the authors should specify which endogenous gene was used as a control and whether the analysis method was delta CT or not. These details should be clearly stated in the text. "
Response 3: Thank you for your feedback. The information regarding the endogenous gene and the analysis method (2-ΔΔCT) were provided at the end of the page 12, highlight in grey:
"Relative gene expression was calculated using the 2-ΔΔCT method. All data were normalised to the gene TBP. TBP was used as endogenous control as it showed the most stable mRNA expression among all tested housekeeping genes (36B4, Ppib, TFIIB), exhibiting the lowest coefficient of variation across all samples."
However, these information may have gone unnoticed in the previous manuscript, because the paragraph was cut by the figure 5 (protocol design). Therefore, we have moved this paragraph to appear before Figure 5, in order to improve clarity and visibility in the revised manuscript.
Reviewer 2 Report
Comments and Suggestions for Authors
The results are interesting to this reviewer. The paper would be improved if the background static magnetic field measurements were included. Additionally it would help to indicate if the applied signals are on the carrier frequencies for the 2G and 5G signals. There is significant data including data from our lab showing that changes of a micro-Tesla can modify the biological response to weak radio frequency fields.
A side comment. My wife had her computer located close to our WiFi transmitter and complained of cold feet after using it for several hours before dinner. We removed the WIFI and went to a wired connection without telling her and in 3 days she no longer made any complaints on cold feet.
Author Response
Comments 1:"The paper would be improved if the background static magnetic field measurements were included."
Response 1: The distance between the animals and the antennas is about 80 cm. Since this distance is much larger than the wavelength at 0.9 GHz and at 3.5 GHz, the field can be characterized by its electric field component at each of these two frequencies. Furthermore, there is no static magnetic field produces by the radiating device. However, we forget to mention the distance between the animals and the antennas. We added two sentence:
- pages 11-12, highlight in green "Antennas were aligned horizontally in the climatic chamber, 80 cm above the exposed rats’ boxes, thus at a height larger than 2.4λ and 9λ, at 900 MHz and 3.5 GHz, respectively, with λ for the wavelength. The position of the antennas is adjusted to minimize the variation of the field amplitude within each cage and between cages."
- page 12, highlight in green "The transmitting device does not generate a static magnetic field."
Comments 2: "Additionally it would help to indicate if the applied signals are on the carrier frequencies for the 2G and 5G signals."
Response 2: The carrier frequency for 2G signals is 900 MHz and for 5G signals is 3.5 GHz. To ensure accuracy in our exposure conditions, we verified that the applied RF signals were within the expected carrier frequency using an Emespy exposimeter.
Comments 3: "A side comment. My wife had her computer located close to our WiFi transmitter and complained of cold feet after using it for several hours before dinner. We removed the WIFI and went to a wired connection without telling her and in 3 days she no longer made any complaints on cold feet."
Response 3: Thank you for your comments. It was not the first, many people report this feeling and we hope to conduct a study to confirm this observation, especially on the hands.